# The association between menstrual hygiene, workplace sanitation practices and self-reported urogenital symptoms in a cross-sectional survey of women working in Mukono District, Uganda

**Sarah A. Borg**[1], **Justine N. Bukenya**[2], **Simon P. S. Kibira**[2], **Petranilla Nakamya**[2], **Fredrick E. Makumbi**[3], **Natalie G. Exum**[4], **Kellogg J. Schwab**[4], **Julie Hennegan**[1,4]*

1 Maternal, Child and Adolescent Health Program, Burnet Institute, Melbourne, VIC, Australia, 2 Department of Community Health and Behavioural Sciences, School of Public Health, College of Health Sciences, Makerere University, Kampala, Uganda, 3 Department of Epidemiology and Biostatistics, School of Public Health, College of Health Sciences, Makerere University, Kampala, Uganda, 4 The Water Institute, Department of Environmental Health and Engineering, Johns Hopkins Bloomberg School of Public Health, Baltimore, MD, United States of America

* julie.hennegan@burnet.edu.au

## Abstract

### Background

Women worldwide experience challenges managing their periods. Menstrual and genital hygiene behaviours have been linked to negative health outcomes, including urogenital symptoms and confirmed infections. However, evidence testing this association has been limited and mixed. This study aimed to (1) describe the menstrual care practices and prevalence of self-reported urogenital symptoms among working women in Mukono District, Uganda, and (2) test the associations between menstrual and genital care practices, and urogenital symptoms.

### Methodology

We undertook a cross-sectional survey of women aged 18–45 working in markets, schools, and healthcare facilities in Mukono District, with 499 participants who had menstruated in the past two months included in this analysis. We developed an aggregated measure of menstrual material cleanliness, incorporating material type and laundering practices. Associations with urogenital symptoms were tested using the aggregated material cleanliness measure alongside the frequency of changing materials, handwashing before menstrual tasks, and sanitation practices.

### Results

Among our sample, 41% experienced urogenital symptoms in the past month. Compared to women exclusively using disposable pads, using appropriately cleaned or non-reused

**Data Availability Statement:** Deidentified data relevant to the analysis presented in this

publication are available from the Open Science Framework repository (https://osf.io/nzjtq/).

**Funding:** This study was funded by the Osprey Foundation of Maryland (http://www.ospreyfdn.org/ received by KS, no grant no). JH is supported by a NHMRC Investigator Grant (https://www.nhmrc.gov.au/ GNT2008600) and the Reckitt Global Hygiene Institute (RGHI) (https://rghi.org/). The views expressed are those of the authors and not necessarily those of RGHI. The authors gratefully acknowledge the contribution to this work of the Victorian Operational Infrastructure Support Program received by the Burnet Institute (https://www.burnet.edu.au/). The funders had no role in study design, data collection, analysis, decision to publish, or preparation of the manuscript.

**Competing interests:** The authors have declared that no competing interests exist.

improvised materials (PR = 1.33, 95%CI 1.04–1.71), or inadequately cleaned materials (improvised or commercially produced reusable pads) (PR = 1.84, 95%CI 1.46–3.42) was associated with an increased prevalence of self-reported urogenital symptoms in the last month. There was no difference between those using disposable pads and those using clean reusable pads (PR = 0.98; 95%CI 0.66–1.57). Infrequent handwashing before changing materials (PR 1.18, 95%CI: 0.96–1.47), and delaying urination at work (PR = 1.37, 95% CI: 1.08–1.73) were associated with an increased prevalence of self-reported symptoms.

## Conclusion

Prevalence of self-reported urogenital symptoms was associated with the type and cleanliness of menstrual material used as well as infrequent handwashing and urinary restriction. There is a need for interventions to enable women to maintain cleanliness of their menstrual materials and meet their menstruation, urination and hand washing needs at home and work.

## Introduction

As defined in 2021, "Menstrual health is a state of complete physical, mental, and social well-being and not merely the absence of disease or infirmity, in relation to the menstrual cycle" [1]. Women, girls, and other people who menstruate worldwide experience challenges to their menstrual health, resulting from inadequate access to clean menstrual materials, supportive infrastructure for menstrual management, access to knowledge and support, and sociocultural environments which frequently stigmatize menstruation [2].

Menstrual hygiene, the hygienic management of menstrual bleeding, represents one key requirement for menstrual health, describing the use of sufficient, clean menstrual materials and access to supportive facilities for washing the body and laundering materials [2–6]. Menstrual hygiene has been hypothesised as a risk factor for urogenital infections, including reproductive tract infections (RTIs) and urinary tract infections (UTIs). Non-sexually transmitted RTIs with hypothesised links to menstrual hygiene include bacterial vaginosis (BV) and vulvo-vaginal candidiasis (VVC) [7].

BV and VVC can cause significant vaginal irritation, malodour, impact on sex life, self-esteem, and mood disorders [8, 9], and have been associated with an increased risk of HIV infection [10]. BV has been associated with human papillomavirus infection [10] and adverse pregnancy outcomes] 10] including, premature rupture of membranes, preterm delivery, low birth weight, chorioamnionitis, and spontaneous abortion [11]. Menstrual hygiene has also been hypothesised to be linked with UTIs. UTIs can cause significant discomfort, have a detrimental influence on quality of life [12], and complications of UTI, like pyelonephritis, are associated with a significant burden of care due to risk of hospitalisation [13]. Moreover, a qualitative investigation of women's menstrual experiences at work in Uganda found significant worries and discomfort related to urogenital symptoms [14].

While menstrual hygiene practices have hypothesised links with urogenital infection, limited studies have evaluated this relationship, and there have been mixed results [2, 15, 16]. In a 2013 systematic review [10], menstrual hygiene practices (including use of reusable or improvised absorbents like cloth or toilet paper, or a low hygiene index score), were found to be associated with self-reported vaginal discharge, or clinically- or laboratory-confirmed BV in seven

cross-sectional studies [17–23]. However, there was significant variation in methodology and overall low quality [10]. Meta-analysis of a subset of studies found no association between clinically confirmed BV and menstrual hygiene [10]. Since this review, further cross-sectional studies have found that urogenital symptoms or clinically- or laboratory-confirmed RTIs or UTIs may be more likely in women with a number of different menstrual hygiene practices including type of material used, frequency of absorbent material changing, washing and drying practices, handwashing, and availability of private sanitation facilities and soap and/or water [24–29]. A 2015 hospital-based case-control study in Odisha, India found increased likelihood of symptoms or diagnosis of urogenital infection with "less adequate" menstrual practices, including a lack of privacy for changing, cleaning and washing during menstruation [30]. Symptoms of, and laboratory-confirmed, UTIs and BV were also more likely in women using reusable cloth compared to using disposable pads [30]. A 2020 school-based interventional prospective cohort study in Rwanda found no difference in rates of laboratory-confirmed UTI but a decreased rate of vulvovaginal symptoms (bothersome discharge and/or odour) in users of menstrual pads compared to no menstrual pads [31]. A 2022 study nested within a pair-matched cohort study in Odisha, India found that household latrines or bathing areas with access to piped water had a moderate impact on adequate menstrual hygiene practices (adequate frequency of absorbent change, washing the body with soap, and privacy for managing menstruation) but no evidence of effect on self-reported urogenital infection symptoms [32]. A recent 2022 systematic review [33] found several menstrual hygiene practices that increased risk of laboratory-confirmed RTIs, including increased risk of VVC with use of reusable materials, drying reusable pads inside the house and storing them inside the toilet. Bathing with water alone showed an increased association with symptoms of urogenital infection compared with washing with soap and water during menstruation. Not drying the genital area or using cloth for drying it, and not handwashing, were associated with a higher risk of genital infections. In summary, extant approaches to testing the association between menstrual hygiene practices and urogenital infection and findings have been mixed. Many studies have tested hygiene practices individually, rather than in combination. Sanitation facility type as a variable denoting potential exposure to fecal contamination of fomites within the environment [34] in which menstrual materials are changed has not been specifically explored in the previous literature. Further, most past research has drawn a dichotomy between disposable pads and other menstrual materials. The availability and use of commercially produced reusable products is increasing. This necessitates an updated consideration of menstrual materials and practices, including a meaningful comparison of disposable and reusable materials. The disposal of single-use menstrual products presents a significant challenge for waste management and global objectives for sustainable consumption [35]. Reusable products can offer a waste and cost-effective alternative, however concerns of infection associated with inadequate washing and drying have attenuated enthusiasm for these products in low-and-middle-income country contexts. Research is needed to understand hygiene practices associated with reusable menstrual products and links to urogenital infection and irritation.

Limited research has described adult women's menstrual practices, particularly in the workplace. As the importance of menstrual health has gained increased recognition, most focus in research and intervention has been for adolescents [36]. Indeed, past research investigating linkages between hygiene practices and urogenital infection have focused on home-based behaviours or have not incorporated measures of workplace practices. Working women are likely to spend a significant amount of time in their workplace, and these experiences and related needs require further exploration and understanding to be able to identify avenues for intervention [37]. Understanding the menstrual practices of women in workplaces, alongside challenges for urination, which may increase risk of UTI [38], would allow a better

understanding of potential causes of genital irritation in working women and inform improved support for menstruation in the workplace.

## The present study

This study aimed to describe the menstrual care practices of working women in Mukono District, Uganda, to report the prevalence of self-reported urogenital symptoms among this population, and to investigate the associations between menstrual care practices, urinary restriction, and the prevalence of self-reported symptoms of urogenital infections (BV, VVC and UTI). Our study was undertaken as part of a broader exploration of women's experiences of menstruation and sanitation in the workplace. This research focused primarily on women working in markets; a common worker group among women in Mukono with varied access to workplace water, sanitation and hygiene (WASH). Further we included a small proportion of teachers and health care facility workers; groups for which school and health care facility WASH is a recognised data and service gap [39]. All three groups represent workplaces for which government has responsibility for service provision and would be able to implement changes in response to research findings.

Extending on past research, this study aimed to disrupt the dichotomy of disposable pads compared to other methods and compare commercially produced reusable pads and improvised reusable materials, as well as compare clean and unclean reusable materials. We developed an aggregated measure of material cleanliness based on past research as an exposure measure. In addition, we investigated other key practices, including frequency of changing materials, handwashing before menstrual tasks, and sanitation facility types and practices, like availability of water for genital washing. Past qualitative research among the study population [14] found that almost all participants believed it was essential to wash their genitals every time they changed their menstrual materials, thus representing a key hygiene practice accompanying menstruation.

## Material and methods

### Study design

This cross-sectional study formed part of a larger mixed-methods research program that explored sanitation and menstrual experiences of women working in Mukono District, Uganda [14, 40]. This paper reports findings from a cross-sectional survey.

### Study context

Mukono District is located in central Uganda. The projected population was 701,400 in 2020, with females making up 51.6%, and approximately 73% of the total population living in rural areas [41, 42]. However, some women working in Mukono District reside in the surrounding districts.

According to the 2017 Performance Monitoring and Accountability survey [43], only 35% of women in Uganda reported having everything they need to manage their menstruation. In the same survey, most women reported using disposable pads (84% urban, 59% rural), followed by reusable cloths (22% urban, 49% rural) as menstrual absorbents.

Household basic hygiene coverage (availability of a handwashing facility with soap and water at home [44]) in Uganda has increased from 2.6% in 2000 (1.5% rural, 8.8% urban) to 22.6% in 2020 (18.2% rural, 35.8% urban) [45]. In Central 2 Region, where Mukono District is located, this has increased from 11.3% in 2011 to 40.2% in 2016 [45]. At least basic sanitation in households (use of improved facilities (including flush/pour flush toilets connected to piped

sewer systems, septic tanks or pit latrines, pit latrines with slabs (including ventilated pit latrines), and composting toilets [44] that are not shared with other households) has increased from 16.7% in 2000 (14.5% rural, 28.9% urban) to 19.8% in 2020 (17.1% rural, 28.0% urban) [45]. In Central 2 Region, this increased from 30.4% in 2009 to 36.9% in 2016 [45].

### Recruitment

All marketplaces in Mukono District that operated for at least three days a week for a minimum of eight hours a day were identified with assistance from the local government and included (n = 10). Estimated female worker population of most markets ranged from 10–150, with one large central market hosting an estimated 950 female workers. Neighbouring government primary schools and public healthcare facilities were then sampled to support feasibility. For each of the ten markets, five teachers and five healthcare facility workers were recruited.

In each market, 50% of the female worker population were sampled, except for the largest market in the area, for which 20% of the market population was sampled to ensure adequate participant representation from the smaller markets. Every second or fifth working woman was systematically sampled by enumerators who mapped each market. Women aged 18–45 years, who had worked at least three days per week over the last month at their relevant workplace were eligible [14], with ineligible participants replaced by a neighbouring worker. Women who reported menstruating in the past six months were asked questions related to menstruation. The larger quantitative study results detailing participants' broader menstrual health needs at work and explorations of associations between unmet needs and women's work and wellbeing are published elsewhere [40]. In this study, we used data from participants who reported menstruating in the last two months to align recent menstrual hygiene practices with reports of urogenital symptoms in the past month.

### Data collection

Survey data was collected in March 2020. Participants selected whether surveys were delivered in Luganda or English by ten experienced female Ugandan enumerators who had received five days of training. Written informed consent was required for participation and participants were made aware of their right to decline to answer any questions. Surveys lasted between 45 and 60 minutes. Participants were given a bar of soap (approximately US$1) as a thank you for participation.

### Measures

Survey questions were developed in English and informed by existing evidence [46, 47] and findings from a qualitative phase of research, reported previously [14]. Cognitive interview testing was undertaken to refine the topics included and questions used [48]. Questions were designed to capture aspects across the integrated model of menstrual experience [36], including menstrual health needs, experiences, and consequences.

Questions from the Menstrual Practices Questionnaire [49] were used to collect information about menstrual hygiene practices, such as type of menstrual materials used and how they are disposed. Questions captured practices at home and in the workplace and were used to describe women's menstrual care and challenges at work, and to construct exposures.

Urogenital symptoms were measured by asking participants if they had experienced one or more self-reported symptoms of urogenital infection (burning or discomfort when urinating; itching or burning in the genital region; unpleasant or fishy odour from genital area; or abnormal vaginal discharge (unusual texture and colour: e.g., milky, white, grey, green, or yellow discharge)) in the last month.

Data on covariates was collected using sociodemographic questions capturing participants' age, level of educational attainment, and workplace type. A 5-item lived poverty index [50] asking how frequently participants' households had gone without food, water, medical care, cooking fuel, or cash income in the past year was used to calculate a poverty score of 0–20 for each participant, with 20 denoting the highest poverty score.

## Construction of variables

Menstrual hygiene practices investigated for their association with urogenital symptoms in past studies were reviewed to inform the variables included and constructed for this study [10, 17–33]. A full list of questions used in this study are presented in S1 Table in S1 File.

**Menstrual absorbent cleanliness.** We developed an aggregated measure of absorbent cleanliness, incorporating material type, along with washing and drying practices. Commercially produced single-use (disposable) pads were considered the cleanest material and reference group. Reusable menstrual pads that had been appropriately cleaned were constructed as the next group. Reused materials were considered clean if they were washed or soaked with soap or detergent and completely dried every time before use during the last period. Improvised methods, including cloth/towel, underwear alone, cotton wool, gauze (medical), and/or toilet paper, were then grouped. Those that were used only once (that is, where participants had reported that they did not wash and reuse any materials) or were cleaned appropriately, were grouped as 'improvised materials clean or not reused'. Finally, any reused materials (commercially produced or improvised) that were not cleaned appropriately were included in the final grouping. Only six participants reported using reusable pads that were "not clean", and thus this group was combined with improvised materials that were "not clean". When reporting the type of menstrual material used, participants reported all the materials they had used during their last period. Participant material was grouped according to the 'least clean' category of materials. This variable construction allowed all menstrual material types to be included in the same final model.

**Washing and drying practices.** Sub-group analysis assessing individual material cleaning practices only for those reusing materials were included for comparison. In addition to washing with soap or detergent and drying completely, additional cleaning methods were assessed; covering materials while drying, drying in the sun, and ironing materials before reuse.

**Genital washing.** The location usually used to urinate, sanitation facility type usually used for urinating when at work (pour flush toilet, ventilated improved pit latrine or composting toilet, pit latrine with a slab, unimproved facility (pit latrine without a slab, bucket/pan, bushes/waterway, in the corridors, or bathroom) or no facility, and never urinates at work), and whether water was available for washing at that location was used to capture the potential exposures to fecal contamination of surfaces that may be touched prior to changing menstrual materials, and to unclean water for genital washing during menstruation. The qualitative portion of the study found that almost all participants believed it was essential to wash their genitals every time they changed their menstrual materials [14]. It was therefore assumed that participants washed their genitals each time their menstrual materials were changed. Questions about the place usually used for urination when at work and whether water for washing was available and/or if participants brought their own water for cleansing were used as a proxy for location and water used for genital washing when changing menstrual materials.

**Menstrual hygiene practices.** Other menstrual hygiene practices included the frequency of changing menstrual materials during the heaviest day of the period. Participants reported perceptions of being able to change as often as they wanted to when at home or work, assessed using single items from the Menstrual Practice Needs Scale [48, 51].

**Outcomes: Urogenital symptoms.** Potential clusters of self-reported symptom types were investigated, however, no apparent patterns emerged in groupings of different symptoms experienced by participants. Thus, the outcome of self-reported symptoms was grouped and analysed as a binary variable of either experienced or did not experience symptoms.

## Statistical analysis

All analyses were performed using Stata SE version 17 (STATA Corp., Texas, USA).

The associations between menstrual hygiene and urination practices with self-reported urogenital symptoms were tested using Poisson regression models with robust variance and standard errors. Binary associations between menstrual hygiene and urination practices with urogenital symptoms were assessed. A multivariable Poisson regression model was used to test associations between self-reported symptoms and all the response variables that yielded statistically significant results in the independent regression models. This model was adjusted for likely sociodemographic confounders including age, poverty score, educational attainment, and workplace type. The Poisson model results were presented as adjusted prevalence ratios (PR) with 95% confidence intervals (CI).

A sub-group analysis of participants who washed and reused materials is presented to disaggregate material cleaning practices and describe associations with urogenital symptoms. Groups within this sub-sample were too small for further adjustment.

## Ethical approvals

The Johns Hopkins Bloomberg School of Public Health Institutional Review Board (IRB: 00010015) and Makerere University School of Public Health Higher Degrees, Research and Ethics Committee (HDREC: 739) provided ethical approval. The study was approved by the Uganda National Council for Science and Technology (UNCST) (ref: SS 5143). Recruitment of participants from their workplaces was permitted by workplace administrators (Headteachers, Healthcare Facility Administrators, and Market Chairpersons). The Mukono district chief administrator's office and the Mukono Municipality Town Clerk's Office also provided approval for the study in the area. Participants who reported symptoms were referred to nearby public health facilities for further management. No identifying information was collected linked to survey data.

## Results

### Participant characteristics

600 women aged 18–45 years old (mean age = 31, standard deviation (SD) = 7.4) working in Mukono District, Uganda participated in the Women and Workplaces Quantitative Survey. Women were included in the present analysis if they reported menstruating in the last two months (n = 499). Participant characteristics are presented in Table 1.

### Menstrual care at home and at work

Participants used varying combinations of materials to manage their periods at home and at work (Table 2), with the majority (64%, n = 319) using disposable pads exclusively. A detailed breakdown of the menstrual materials used is presented in S2 Table in S1 File with a breakdown of material cleanliness in S3 Table in S1 File.

A total of 9% of participants (n = 45) reported using a commercially produced reusable product. Of these, eight percent (n = 39) used only reusable pads (n = 25) or these products combined with disposable pads (n = 13) or with an improvised method (n = 1) and cleaned

**Table 1. Participant characteristics.**

| Participant characteristic s | % (n) |
|---|---|
| **Age** | |
| 18–24 years old | 25.1 (125) |
| 25–29 years old | 22.7 (113) |
| 30–39 years old | 34.9 (174) |
| 40–45 years old | 17.4 (87) |
| **Workplace type** | |
| Marketplace | 82.4 (411) |
| Healthcare facility | 8.8 (44) |
| School | 8.8 (44) |
| **Highest level of school attended** | |
| Never attended | 4.2 (21) |
| Primary school | 31.7 (158) |
| Secondary school | 46.1 (230) |
| Post-secondary technical | 12.6 (63) |
| Post-secondary university | 5.2 (26) |
| Post-graduate | 0.2 (1) |
| **Poverty score (min: 0 max: 20)** | |
| Mean | 4.4 (SD = 3.7) |
| **Self-reported general health** | |
| Very good | 9.6 (48) |
| Good | 76.6 (382) |
| Bad | 12.4 (62) |
| Very bad | 1.4 (7) |

SD = Standard deviation

them appropriately. Two percent of participants (n = 6) used designated reusable pad products that were unclean, these were categorised as 'reused methods not clean' in the same category as unclean reused improvised materials.

Over a quarter of participants (27%, n = 136) used an improvised menstrual material. Most cleaned these appropriately, while four percent (n = 22) reused unclean improvised materials.

The locations that materials were most often changed and disposed of at home and at work are detailed in Table 2. Sixty-five percent of participants (n = 324) were always able to change their materials when they wanted to, and the majority (51%, n = 252) changed their materials three times a day on the heaviest day of their last menstrual period. Less than half (41%, n = 202) washed their hands every time before changing their materials. Eleven percent (n = 57) usually went home to urinate when at work, 53% (n = 263) used a sanitation facility that had water for washing available, 27% (n = 136) used a sanitation facility, bucket/pan or bushes/waterway and brought their own water for cleansing, and eight percent (n = 41) used a sanitation facility, bucket/pan or bushes/waterway and did not have water available for washing nor bring their own water for cleansing. The majority (86%, n = 431) usually used an improved facility to urinate at work, 8% (n = 38) usually used an unimproved facility, 4% (n = 20) used no facility, and 2% (n = 10) went home to urinate when at work or never needed to urinate at work.

Twenty-eight percent of participants (n = 141) washed and reused menstrual materials. Of these, the vast majority (94%, n = 132) used soap or detergent to soak or wash their materials every time, 24% (n = 34) ironed their materials before using them, and 12% (n = 17) dried

**Table 2. Menstrual materials used and menstrual care practices at home and work during last period or last month.**

| Materials used | | % (n) |
|---|---|---|
| Disposable pad | | 63.9 (319) |
| Reusable pads clean* or not reused *(used with or without disposable pads)* | | 7.6 (38) |
| Improvised methods[†] clean* or not reused *(used with or without disposable/reusable pads)* | | 22.8 (114) |
| Reused methods not clean* *(improvised methods[†] (n = 22) and reusable pads (n = 6))* | | 5.6 (28) |
| **Location materials were most often changed when at home** | **% (n)** | **Location materials were most often changed when at work** | |
| Bedroom | 42.3 (211) | Never changed at work or goes home to change | 26.6 (132) |
| Toilet | 6.0 (30) | Toilet | 32.2 (160) |
| Pit latrine | 19.8 (99) | Pit latrine | 29.6 (147) |
| Bathroom | 31.5 (157) | Bathroom | 5.2 (26) |
| Outside/bush | 0.4 (2) | Another area | 6.4 (32) |
| **Where materials were disposed of at home** | **% (n)** | **Where materials were disposed of at work** | |
| Did not dispose of any materials | 22.2 (111) | Did not dispose of any materials at work or took home to dispose | 25 (90) |
| Pit latrine or toilet | 68.5 (342) | Pit latrine or toilet | 43 (156) |
| Burned | 4.2 (21) | Burned | 1 (4) |
| Household or community rubbish | 3.6 (18) | Bin/bucket inside sanitation place | 29 (105) |
| Bush/buried or other | 1.4 (7) | Bush/buried or bin/bucket outside sanitation place | 2 (8) |

| Able to change menstrual materials when wanted to at home and at work | |
|---|---|
| Always at home & always at work (or did not need to change during workday or did not attend work during period) | 64.9 (324) |
| Sometimes or never at home and/or work | 35.1 (175) |
| **How many times menstrual materials were changed on the heaviest day of last menstrual period** | |
| 1–2 times | 31.3 (156) |
| 3 times | 50.6 (252) |
| 4 or more times | 18.1 (90) |
| **How often participants wash their hands before changing menstrual materials during last period** | |
| Washed hands every time before changing materials during last period | 40.6 (202) |
| Washed hands sometimes or never before changing materials during last period | 59.4 (296) |
| **The place usually used to urinate when at work and whether water for washing is available at the location and/or if participants bring own water for cleansing** | |
| When at work usually goes home to urinate | 11.4 (57) |
| When at work usually urinates at a sanitation facility with water for washing available | 52.7 (263) |
| When at work usually urinates at a sanitation facility or in a bucket/pan or bushes/waterway and brings own water for cleansing | 27.3 (136) |

*(Continued)*

**Table 2.** (Continued)

| Materials used | % (n) |
|---|---|
| When at work usually urinates at a sanitation facility or in a bucket/pan or bushes/waterway with no water for washing available and does not bring own water for cleansing | 8.2 (41) |
| **Type of sanitation facility usually used to urinate when at work** | |
| Pour flush toilet | 38.3 (191) |
| Ventilated improved pit latrine or composting toilet | 21.6 (108) |
| Pit with slab | 26.5 (132) |
| Unimproved or no facility | 11.6 (58) |
| Never urinates at work | 2.0 (10) |
| **Washed and reused any menstrual materials during last period** | |
| Washed and reused any menstrual materials during last period | 28.3 (141) |
| Did not wash and reuse any menstrual materials during last period | 71.7 (358) |
| **Used soap or detergent to soak or wash menstrual materials during last period** | |
| Used soap or detergent to soak or wash materials during last period every time | 93.6 (132) |
| Used soap or detergent to soak or wash materials during last period sometimes or never | 6.4 (9) |
| **Ironed menstrual materials before reusing them during last period** | |
| Ironed materials before using them during last period | 24.1 (34) |
| Dried materials in sun during last period sometimes or never | 75.9 (107) |
| **Dried materials in the sun during last period** | |
| Dried materials in sun during last period every time | 12.1 (17) |
| Dried materials in sun during last period sometimes or never | 87.9 (124) |
| **Menstrual materials were completely dry before using them during last period** | |
| Materials were completely dry every time before using them during last period | 84.4 (119) |
| Materials were completely dry sometimes or never before using them during last period | 15.6 (22) |
| **Menstrual materials you covered with anything when drying** | |
| Materials were not covered while drying during last period | 52.1 (73) |
| Materials were covered while drying during last period | 47.9 (67) |

*Clean: used soap or detergent to soak or wash materials every time and materials were completely dry before using every time in the last month if washed and reused any menstrual materials during last period

†Improvised methods: cloth/towel, underwear alone, cotton wool, gauze, and/or toilet paper

their materials in the sun every time. For the majority (84%, n = 119), their materials were completely dry every time before using them, and just over half (52%, n = 73) did not cover their materials with anything while drying them.

The greatest proportion of participants (43%, n = 156) disposed of their used menstrual materials into a pit latrine or toilet in the workplace, while 29% (n = 105) disposed into a waste

bin or bucket at the sanitation facility, and 25% (n = 132) took materials home with them to dispose of.

## Self-reported urogenital symptoms experienced by participants

Forty-one percent of participants (n = 206) self-reported experiencing one or more urogenital symptom (burning or discomfort when urinating; itching or burning in the genital region; unpleasant or fishy odour from genital area; or abnormal vaginal discharge) in the last month, and 66% (n = 100) discussed their symptoms with a healthcare provider when they occurred. No consistent evidence of patterns among symptom types experienced by participants in the last month emerged (displayed in S4 Table in S1 File).

## Associations between menstrual hygiene and urinary restriction and urogenital symptoms

Associations between menstrual hygiene, urinary restriction, and experiencing one or more urogenital symptom(s) in the last month are displayed in Table 3. Sub-group analysis for participants reusing materials, reporting individual cleaning practices used, and additional cleaning including drying materials in the sun, uncovered, or using an iron on materials, and their binary associations with urogenital symptoms are displayed in Table 4.

**Materials used.** There was no difference in the prevalence of self-reported urogenital symptoms in the last month between participants who used disposable pads only and participants who used reusable pads that were clean or not reused (used with or without disposable pads) (aPR = 0.98, 95% CI: 0.63–1.59; PR = 0.98, 95% CI: 0.66–1.57) during their last period. The prevalence of self-reported urogenital symptoms in the last month was 1.33 times higher (95% CI: 1.04–1.71; PR = 1.51, 95% CI: 1.20–1.90) among participants who used improvised materials that were clean or not reused (used with or without disposable pads and/or reusable pads), and 1.84 times higher (95% CI: 1.46–3.42; PR = 2.26, 95% CI:1.77–2.89) among participants who used reused, unclean materials (improvised materials (n = 22) or reusable pads (n = 6)), than in participants who used disposable pads only during their last period. In the sub-analysis of participants who washed and reused materials (n = 141) (Table 4), the prevalence of self-reported urogenital symptoms in the last month was 1.40 times (95% CI: 1.13–1.71) higher in participants who washed and reused menstrual materials than in participants who did not reuse any during their last period. The prevalence of self-reported urogenital symptoms in the last month was 1.31 times higher (95% CI 0.81–2.15) in participants who did not use soap or detergent to soak or wash their materials every time, 1.13 times higher (95% CI 0.76–1.69) in participants who did not iron their materials before using them, 1.53 times higher (95% CI 0.78–2.98) in participants who did not dry their materials in the sun every time, 1.77 times higher (95% CI 1.34–2.34) in participants whose materials were not completely dry every time before use, and 1.06 times higher (95% CI 0.77–1.46) in participants whose materials were covered while drying during their last period.

**Frequency of changing materials.** The prevalence of self-reported urogenital symptoms in the last month was 1.11 times higher (95% CI 0.89–1.37) among participants who were not always able to change their menstrual materials when they wanted to (or did not need to change during workday or did not attend work during their last period) than in participants who were always able to change their materials when they wanted to. The prevalence of self-reported urogenital symptoms increased along with the number of times menstrual materials were changed (Table 3). However, these findings were not statistically significant.

**Sanitation facility type and hand and genital washing.** The prevalence of self-reported urogenital symptoms in the last month was 1.18 times higher (95% CI: 0.96–1.47; PR = 1.23,

**Table 3. The prevalence of self-reported urogenital symptoms in the past month according to menstrual hygiene and urinary restriction practices.**

| Materials used | % (n) | % of method users who experienced symptoms (n) | % of users who didn't experience symptoms (n) | PR | 95% CI | Adjusted PR (aPR)$^{\alpha}$ | 95% CI |
|---|---|---|---|---|---|---|---|
| Disposable pad | 63.9 (319) | 34.8 (111) | 65.2 (208) | | | | |
| Reusable pads clean* or not reused (used with or without disposable pads) | 7.6 (38) | 34.2 (13) | 65.8 (25) | 0.98 | 0.66–1.57 | 0.98 | 0.63–1.59 |
| Improvised† methods clean* or not reused (used with or without disposable/reusable pads) | 22.8 (114) | 52.6 (60) | 47.4 (54) | 1.51 | 1.20–1.90 | 1.33 | 1.04–1.71 |
| Reused methods not clean* (improvised† methods (n = 22) or reusable pads (n = 6)) | 5.6 (28) | 78.6 (22) | 21.4 (6) | 2.26 | 1.77–2.89 | 1.84 | 1.46–3.42 |
| **Urinary restriction** | **% (n)** | **% of those who undertook activity & experienced symptoms (n)** | **% of those who undertook activity & didn't experience symptoms (n)** | **PR** | **95% CI** | | |
| Needed to delay urinating at work in the last month sometimes or always | 60.9 (304) | 47.4 (144) | 52.6 (160) | 1.49 | 1.17–1.89 | 1.37 | 1.08–1.73 |
| Never needed to delay urinating at work in the last month never | 39.1 (195) | 31.8 (62) | 68.2 (133) | | | | |
| **Menstrual hygiene practices** | | | | | | | |
| Able to change menstrual materials when wanted to always at home & always at work (or did not need to change during workday or did not attend work during period) | 64.9 (324) | 39.8 (129) | 60.2 (195) | | | | |
| Able to change menstrual materials when wanted to sometimes or never at home and/or work | 35.1 (175) | 44 (77) | 56.0 (98) | 1.11 | 0.89–1.37 | | |
| Changed menstrual materials 1–2 times on the on the heaviest day during last period | 31.3 (156) | 37.2 (58) | 62.8 (98) | | | | |
| Changed menstrual materials 3 times on the on the heaviest day during last period | 50.6 (252) | 40.9 (103) | 59.1 (149) | 1.10 | 0.85–1.42 | | |
| Changed menstrual materials 4 or more times on the on the heaviest day during last period | 18.1 (90) | 50.0 (45) | 50.0 (45) | 1.34 | 1.01–1.80 | | |
| Washed hands every time before changing materials during last period | 40.6 (202) | 36.1 (73) | 63.9 (129) | | | | |
| Washed hands sometimes or never before changing materials during last period | 59.4 (296) | 44.6 (132) | 55.4 (164) | 1.23 | 0.99–1.54 | 1.18 | 0.96–1.47 |
| **Genital washing** | | | | | | | |
| When at work usually goes home to urinate | 11.4 (57) | 36.8 (21) | 63.2 (36) | | | | |
| When at work usually urinates at a sanitation facility with water for washing available | 52.7 (263) | 39.9 (105) | 60.1 (158) | 1.08 | 0.75–1.57 | | |
| When at work usually urinates at a sanitation facility or in a bucket/pan or bushes/waterway and brings own water for cleansing | 27.3 (136) | 45.6 (62) | 54.4 (74) | 1.24 | 0.84–1.82 | | |
| When at work usually urinates at a sanitation facility or in a bucket/pan or bushes/waterway with no water for washing available and does not bring own water for cleansing | 8.2 (41) | 43.9 (18) | 56.1 (23) | 1.19 | 0.73–1.94 | | |
| **Sanitation facility** | | | | | | | |
| Pour flush toilet | 38.3 (191) | 38.7% (74) | 61.3 (117) | | | | |
| Ventilated improved pit latrine or composting toilet | 21.6 (108) | 36.1% (39) | 63.9 (69) | 0.93 | 0.68–1.27 | | |
| Pit with slab | 26.5 (132) | 47.0% (62) | 53.0 (70) | 1.21 | 0.94–1.56 | | |
| Unimproved or no facility | 11.6 (58) | 44.8% (26) | 55.2 (32) | 1.16 | 0.82–1.62 | | |

*(Continued)*

**Table 3.** (Continued)

| | | | | | | |
|---|---|---|---|---|---|---|
| Never urinates at work | 2.0 (10) | 50.0% (5) | 50.0 (5) | 1.29 | 0.68–2.46 | |

PR = prevalence ratio; CI = confidence interval

*Clean: used soap or detergent to soak or wash materials every time and materials were completely dry before using every time in the last month if washed and reused any menstrual materials during last period

†Improvised methods: cloth/towel, underwear alone, cotton wool, gauze, and/or toilet paper

αAdjusting for age, poverty score, workplace type, and highest level of education attainment

95% CI:0.99–1.54) among participants who did not wash their hands every time before changing their materials than among participants who washed their hands every time before changing their materials during their last period. The location used for urination, sanitation facility type classification used for urination at work (pour flush toilet, ventilated improved pit latrine or composting toilet, pit latrine with a slab, unimproved facility (pit latrine without a slab, bucket/pan, bushes/waterway, in the corridors, or bathroom) or no facility, or never urinates at work), and water availability for genital washing were not significantly associated with self-reported urogenital symptoms.

**Urinary restriction.** The prevalence of self-reported burning or discomfort when urinating was 1.59 times higher (95% CI: 1.12–2.26) and the prevalence of any self-reported

**Table 4. The prevalence of self-reported urogenital symptoms in the past month according to material washing and drying practices.**

| Reuse of materials | % (n) | % of method users who experienced symptoms (n) | % of users who didn't experience symptoms (n) | PR | 95% CI |
|---|---|---|---|---|---|
| Washed and reused any menstrual materials during last period | 28.3 (141) | 51.8 (73) | 48.2 (68) | 1.40 | 1.13–1.71 |
| Did not wash and reuse any menstrual materials during last period | 71.7 (358) | 37.2 (133) | 62.8 (225) | | |
| **Washing and drying practices (among those reusing materials n = 141):** | | | | | |
| Used soap or detergent to soak or wash materials during last period every time | 93.6 (132) | 50.8 (67) | 49.2 (65) | | |
| Used soap or detergent to soak or wash materials during last period sometimes or never | 6.4 (9) | 66.7 (6) | 33.3 (3) | 1.31 | 0.81–2.15 |
| Ironed materials before using them during last period | 24.1 (34) | 47.1 (16) | 52.9 (18) | | |
| Did not iron materials before using them during last period | 75.9 (107) | 53.3 (57) | 46.7 (50) | 1.13 | 0.76–1.69 |
| Dried materials in sun during last period every time | 12.1 (17) | 35.3 (6) | 64.7 (11) | | |
| Dried materials in sun during last period sometimes or never | 87.9 (124) | 54.0 (67) | 46.0 (57) | 1.53 | 0.78–2.98 |
| Materials were completely dry every time before using them during last period | 84.4 (119) | 46.2 (55) | 53.8 (64) | | |
| Materials were completely dry sometimes or never before using them during last period | 15.6 (22) | 81.8 (18) | 18.2 (4) | 1.77 | 1.34–2.34 |
| Materials were not covered while drying during last period | 52.1 (73) | 50.7 (37) | 49.3 (36) | | |
| Materials were covered while drying during last period | 47.9 (67) | 53.7 (36) | 46.3 (31) | 1.06 | 0.77–1.46 |

PR = prevalence ratio; CI = confidence interval

urogenital symptom in the last month was 1.37 times higher (95% CI: 1.08–1.73; PR = 1.49, 05% CI:1.17–1.89) among participants who sometimes or always needed to delay urinating at work than among participants who never needed to delay urinating at work in the last month.

**Materials used, handwashing, and urinary restriction.**    Multivariable Poisson regression analysis adjusting for age, poverty score, workplace type, and highest level of education attainment (presented as an adjusted prevalence ratio (aPR) in Table 3) looked at associations between self-reported urogenital symptoms and materials used, frequency of handwashing prior to changing materials, and urinary restrictions at work. Analysis showed that compared to using disposable pads only, using clean or not reused improvised materials (with or without disposable and/or reusable pads) (PR = 1.51, 95% CI: 1.20–1.90; aPR = 1.33, 95% CI: 1.04–1.71), or unclean reused materials (either reusable pads or improvised materials, with or without disposable pads) (PR = 2.26, 95% CI: 1.77–2.89; aPR = 1.84, 95% CI: 1.46–3.42) was associated with an increased prevalence of self-reported urogenital symptoms in the last month among participants. Compared to always washing their hands before changing materials, participants who sometimes or never washed their hands before during their last menstrual period (PR = 1.23, 95% CI: 0.99–1.54; aPR = 1.18, 95% CI: 0.96–1.47) had an increased prevalence of self-reported urogenital symptoms in the last month. Compared to participants who never needed to delay urination at work, participants who sometimes or always needed to delay urination at work in the last month (PR = 1.49, 95% CI: 1.17–1.89; aPR = 1.37, 95% CI: 1.08–1.73) had an increased prevalence of self-reported urogenital symptoms in the last month.

## Discussion

This study described the menstrual practices of women working in Mukono District, Uganda, their prevalence of self-reported urogenital symptoms, and the associations between menstrual hygiene practices and urogenital symptoms. Forty-one percent of participants self-reported experiencing one or more urogenital symptom within the past two months. Among this group, two thirds (66%) discussed their symptoms with a health care provider. This finding alone suggests that most women found urogenital symptoms to cause sufficient discomfort and concern to seek care. Further research is needed to understand investigation and management of such symptoms, and quality of care received.

The menstrual hygiene practices that were significantly associated with urogenital symptoms after adjustment for age, poverty score, workplace type, and education, included menstrual absorbent cleanliness, handwashing prior to changing materials, and urinary restriction. Using improvised materials that were cleaned appropriately or not reused was associated with a 33% greater prevalence of symptoms. A hypothesis for this finding is that even if not reused, improvised materials may not be adequately clean on initial use. Irritation from improvised methods may also occur due to chafing, as they are not held in place as easily as designated pad products and are not designed for the purpose of prolonged skin contact. Previous qualitative studies have found reports of irritation and chafing with improvised materials like cloth [52, 53]. Use of inadequately cleaned reusable materials, that is materials that were not washed/soaked with soap/detergent or completely dried, was associated with an 84% greater likelihood of reporting urogenital symptoms. While this represented a small proportion of the sample (6%), this population of women bears a significant burden of discomfort with 79% reporting at least one urogenital symptom.

Most women using reusable menstrual pads appropriately cleaned them, and there was no difference in symptom prevalence among participants using disposable compared to clean reusable pads. This finding contrasts with previous research which has found urogenital infections were more common in women using reusable materials compared to disposable pads [10, 17–23, 30, 33], however, these studies did not differentiate between improvised reusables

and reusable pads. These findings are particularly important in the context of increasing uptake of commercially produced reusable menstrual pads as a cost- and environmentally sustainable alternative to disposable pads. Our results suggest that, in this study setting, women taking up reusable clean pads experience the same benefits of lower prevalence of urogenital symptoms as those using disposables.

Needing to delay urination at work was associated with a significantly greater prevalence of symptoms, particularly burning or discomfort when urinating, a symptom of UTI, which is in line with evidence supporting delayed urination as a risk factor for UTI [38]. Inconsistent handwashing prior to changing menstrual materials showed an association with symptoms in binary and multivariable models, although falling short of statistically significant thresholds. Previous cross-sectional studies have found an association between inadequate or lack of facilities available for handwashing and diagnosed genital infection or urogenital symptoms [27, 28, 54]. Along with providing adequate facilities for urination, handwashing stations that can be used prior to changing menstrual materials may help support genital health.

The prevalence of self-reported symptoms increased with an increased frequency of changing menstrual materials; however, this was not a statistically significant finding. It is plausible that women using poorer quality absorbents need to change them more often. Conversely, past research has found that women diagnosed with BV reported a lower frequency of material changing [24, 30].

Across our sample, many women struggled with menstrual care and sanitation access in their workplace. A total 61% of participants reported needing to delay urinating at work in the past month, and over a third (35%) were not able to change their menstrual materials when they wanted to. Further over four in ten (43%) of participants reported disposing of their used menstrual materials into the workplace sanitation facility, presenting risks to sanitation infrastructure and the environment [35]. Women's menstrual experiences in the workplace have been neglected [55], with only more recent attention highlighting the many challenges faced [14, 56, 57]. More research and action are needed to understand women's menstrual needs at work and identify effective strategies for support.

## Strengths and limitations

An in-depth qualitative study to investigate women's menstrual experiences at work in Uganda [14] and previous research were used to inform development [40] and selection of survey questions for this analysis. Building on past research, this study took a novel approach to assessing associations between menstrual hygiene materials and urogenital symptoms, allowing for meaningful comparison between reusable and disposable materials as well as clean and unclean reusable materials.

Even though we were unable to take a full count of the workers and randomly sample them, a feasible yet rigorous approach of proportional systematic sampling in the marketplaces was undertaken [40]. Due to practical limitations, only a small number of teachers and healthcare facility workers were included. Multivariable comparisons included adjustment for workplace type, poverty, and education level, however, results mostly describe the experiences and associations between practices and urogenital symptoms among women working in markets. Future research is needed to capture the sufficiency of workplace WASH services across worker types in different contexts. Our findings reported here and elsewhere [14, 40] highlight that many women are likely to experience significant menstrual-related challenges in their workplace and that more support is needed.

Participants were not asked about sexual practices to determine risk of sexually transmitted infections (STIs). STIs may have increased the likelihood of urogenital symptoms. There is

likely to be a symptom overlap with STIs and non-sexually transmitted urogenital infections, and sexual intercourse may increase the risk of BV, VVC, and UTI. Women exposed (via skin contact) to water contaminated with the cercariae of schistosomes (also known as "bilharzia") may have experienced urogenital symptoms due to female genital schistosomiasis (FGS), instead of irritation related to menstrual management. This study did not find an association between urination location and genital wash water availability, and urogenital symptoms. However, participants were also not asked about prior exposures to bilharzia-infested waters or about cleanliness of the genital wash water and whether they used soap to wash their genitals. Information about whether water was available (either present at the sanitation facility used for urination, brought by the participant, or neither) when changing menstrual materials, with the assumption made that this was used for genital washing, was included as potential exposure FGS during menstruation. FGS is prevalent in Uganda. The national prevalence of schistosomiasis in females in Uganda in 2016–17 was 24% [58]. FGS occurs in approximately half (33 to 75%) of females with schistosomiasis infection [59] and is one of the most common gynaecologic conditions in Africa [60]. Symptoms of FGS include vaginal discharge and genital itching or burning sensation [61], which, overlap with symptoms of BV, VVC, and UTI. Prevalence of self-reported urogenital symptoms could have been due to STIs or FGS, and not menstruation or urinary management practices.

Answers to questions about menstrual and urinary management and urogenital symptoms are subject to recall and social desirability bias, particularly if participants were embarrassed to report menstrual care practices like reusing unclean materials and infrequent handwashing, and experiencing urogenital symptoms in interviews. Thus, participants may have underreported less hygienic practices and may also have underreported symptoms. This may have dampened the potential effect size.

Because this cross-sectional study measures prevalence, we cannot differentiate aetiology and outcomes, and temporal sequence between exposure and disease cannot be established. Most studies in this field are cross-sectional, with a small number of case-control and cohort studies [30–32, 54, 62]. Many other studies have used self-reported symptoms as the outcome, but some did have clinical confirmation [18–21, 29, 31]. This cross-sectional study similarly only used self-reported symptoms as a measure of urogenital infection. Current Uganda Clinical Guidelines [63] recommend syndromic management. Therefore, presentations of the self-reported symptoms used as outcomes measures in this study would have likely resulted in physical examination and management with antibiotics, without use of diagnostic tests, thus, laboratory confirmation of urogenital infections would not have been feasible in the general population within this study setting. Further, women report distress and discomfort at experiencing urogenital symptoms [8, 9, 13, 14, 64] which are likely to impact on their quality of life, regardless of diagnostic results.

## Conclusions

Working women in markets, healthcare facilities and schools in Mukono, Uganda experience a high burden of urogenital symptoms. Urogenital symptoms represent a significant discomfort, and potential health risk, for women and these findings indicate the need for greater attention to this challenge. Using improvised menstrual materials and inadequately cleaned materials was associated with a greater prevalence of symptoms. A choice of affordable, acceptable, and high-quality menstrual products is needed so that women can use safe methods to meet their individual menstrual needs. The qualitative portion of this study [14] found that women want more information about the performance of different menstrual materials available to them. The findings of this study indicate the need to support women to avoid

urogenital discomfort and enable them to make informed choices about their menstrual care. While the results from this study encouragingly did not find a difference in outcomes when comparing disposable to clean reusable pads, further research comparing products and providing information about risk factors for urogenital infections is required. Reliable access to soap and clean water and private spaces to adequately clean and dry reusable materials is required to support menstrual health. Finally, accessible, clean, and affordable sanitation infrastructure is essential to ensure women can urinate as needed in the workplace and further prevent the burden of urogenital infections.

## Supporting information

**S1 File.**
(DOCX)

## Acknowledgments

We are grateful to the participating workplaces and women who shared their experiences. We thank our skilled enumerator team.

## Author Contributions

**Conceptualization:** Sarah A. Borg, Kellogg J. Schwab, Julie Hennegan.

**Data curation:** Petranilla Nakamya.

**Formal analysis:** Sarah A. Borg, Julie Hennegan.

**Funding acquisition:** Kellogg J. Schwab, Julie Hennegan.

**Investigation:** Sarah A. Borg, Justine N. Bukenya, Simon P. S. Kibira, Petranilla Nakamya, Julie Hennegan.

**Methodology:** Justine N. Bukenya, Simon P. S. Kibira, Fredrick E. Makumbi, Natalie G. Exum, Julie Hennegan.

**Project administration:** Simon P. S. Kibira, Petranilla Nakamya, Julie Hennegan.

**Resources:** Kellogg J. Schwab, Julie Hennegan.

**Supervision:** Justine N. Bukenya, Fredrick E. Makumbi, Kellogg J. Schwab, Julie Hennegan.

**Validation:** Simon P. S. Kibira, Fredrick E. Makumbi, Natalie G. Exum, Kellogg J. Schwab.

**Writing – original draft:** Sarah A. Borg.

**Writing – review & editing:** Justine N. Bukenya, Simon P. S. Kibira, Petranilla Nakamya, Fredrick E. Makumbi, Natalie G. Exum, Kellogg J. Schwab, Julie Hennegan.

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
