## [Decision Letter · Decision Letter 0]

6 Jun 2023

PONE-D-23-10980The association between menstrual hygiene, workplace sanitation practices and self-reported urogenital symptoms in a cross-sectional survey of women working in Mukono District, UgandaPLOS ONE

Dear Dr. Hennegan,

Thank you for submitting your manuscript to PLOS ONE. After careful consideration, we feel that it has merit but does not fully meet PLOS ONE’s publication criteria as it currently stands. Therefore, we invite you to submit a revised version of the manuscript that addresses the points raised during the review process.

Please address both reviewers' minor comments, but otherwise well done for a great manuscript!

We look forward to receiving your revised manuscript.

Kind regards,

Alison Parker

Academic Editor

PLOS ONE

Reviewers' comments:

Reviewer's Responses to Questions

**Comments to the Author**

1. Is the manuscript technically sound, and do the data support the conclusions?

Reviewer #1: Yes

Reviewer #2: Yes

2. Has the statistical analysis been performed appropriately and rigorously? 

Reviewer #1: I Don't Know

Reviewer #2: Yes

3. Have the authors made all data underlying the findings in their manuscript fully available?

Reviewer #1: Yes

Reviewer #2: Yes

4. Is the manuscript presented in an intelligible fashion and written in standard English?

Reviewer #1: Yes

Reviewer #2: Yes

5. Review Comments to the Author

Reviewer #1: Overall, my comments should be seen as minor (even though my writing is major:) ) and with small adjustments and amendments I recommend that this paper is published)

Introduction:

- Reusable pads; I think it would strengthen the rationale for this study further if you bring in the sustainability aspect to the reusable trend (you mention it very briefly in the discussion). Your study actually provides a good argument for the need to see well managed reusable pads as a climate friendly alternative to disposable pads (in terms of related health risks).

-I also think that you could introduce the rationale for focusing on MHM in the work place a bit more – its clear in your previously published papers (which I really enjoyed reading). Self-employed women working low paid jobs in somewhat sanitation and hygiene poor environments (markets) can be seen as particularly vulnerable and important to focus on. ‘MHM poverty’ as a concept could be used. Its interesting that you also include a different category of women (health providers and teachers) – but see my comment about these sub-groups below.

- There is no descriptions of the general WASH standards in the market places. I can only imagine that very few toilets are serving huge numbers of women. And I can also imagine that the status of those WASH facilities are not very high. Same for access to water. Do markets have standpipes? Is water free and accesible?

METHODS:

- I just highly appreciate the break-down of the dichotomy of the reusable/non reusable category, in order to address the perceptions and classifications of reusable materials as being seen as ‘unhygienic’ per default. I think this point could also be further stressed in the papers discussion.

I have no expertise to evaluate the quality of your statistical methods.

RESULTS:

- 66% discussed the symptoms with a health provider; This is in my eyes a VERY high proportion – it would be SO interesting to know if they were prescribed antibiotics (which is often the standard treatment regime as you rightly mention in the discussion), since resistance against antibiotics for UTIs is a huge problem in other low income settings.

- You are not mentioning the quite astonishing finding of the large proportion of women who report that they discard the pads in a sanitation facility! Being forced to discard menstrual materials in toilets is a sign that women do not have other suitable private places to manage their menstrual waste. It’s a huge environmental AND engineering problem that pads are discarded into toilets.

- No separate analysis for teachers or health providers is provided - why not? Its VERY relevant to get insights into status and progress of sanitation and MHM facilities and practices here, since sanitation in health facilities and in schools have been named as two major areas of intervention to support universal access to sanitation and to support MHM. Even though it may not be possible to arrive at statistically significant results for the two small sub-groups of respondents, I believe its relevant to see the results. I suspect that health personnel and teachers may have both better access to sanitation and MHM conditions AND lower frequencies of UTIs due to a high level of health literacy. These findings would lead to observations and discussions about the specific vulnerabilities of private sector, self-employed women, compared to women of higher education, in secure and public sector jobs – there is hardly ANY evidence in this field, so these results would be highly appreciated.

DISCUSSION:

- There is no mentioning or discussion of the descriptive findings of the general lack of access to suitable sanitation and hygiene facilities at the workplace. As a minimum I would like if you could refer to your other papers covering this in more detail. MHM in the workplace is still a hugely under-researched area within MHM research – especially in low income settings. I am well aware that this is not the purpose of this paper, but all findings that shed light on the MHM conditions, situation and problems at women’s workplaces, are relevant to report.

- Linked to my comment under results: I think that the discussion could be improved by adding some reflections about your informant group; Who do they represent? They are a special kind of women, and they work in special types of work places with special types of sanitation and hygiene standards/situations. To which groups and settings can your findings be translated? Hundreds of thousands of women in other low income settings are self-employed, growing some produce at home, selling it at markets, or working as suppliers, middle-men, brokers, and storeowners in similar markets all over the world. I know of similar groups of women in urban Ghana, rural Togo, urban Vietnam, rural and urban Sri Lanka. How are your results relevant beyond the specific ‘Ugandan markets’ context?

- Social desirability bias: Specifically, I think you can hypothesise that women may over report the practice of; drying materials fully in the sun and covering drying materials (two messages which are often heard in MHM promotion messages), and using improved sanitation facilities when at work and bringing water with them for washing when changing. Instead of merely mentioning potential bias, I think you should critically reflect on them, using some of your vast contextual knowledge about Ugandan market places and their structural conditions; Is it really realistic that women can access and afford to use improved sanitation facilities near the markets every time they need to change menstrual materials? From my own experience in urban Ghana, this is not very likely. Long queues, high costs, VERY poor hygienic conditions, and no water or wiping materials are just some of the reasons that women do not choose to use facilities. Also, accessing an improved sanitation facility, does NOT mean that its hygienic and has lower levels of infection risks to the women; fecal materials on the floor and walls, full pits, very strong odors and maggots are just some of the problems reported in many public sanitation facilities in low income settings.

CONCLUSION:

I find that the text here becomes rather generic. I would like to see you specifically refer to your study context (markets, low income setting) and the target groups you have investigated. The conclusion about “Reliable access to soap and clean water and private spaces to adequately clean and dry reusable materials is required” is so unspecific, that its almost true for all of us, always. Focus on the workplace. That’s were your study is new and relevant.

Reviewer #2: Thank you for the opportunity to review this interesting manuscript. The authors provide new insights on the links between menstrual hygiene and sanitation types alongside the use of clean/unclean disposable/reusable menstrual/improvised materials with urogenital symptoms. It is particularly useful that they have differentiated between reusable pads and improvised materials.

The authors describe a robust study and thoroughly investigates the limitations of the work. I have made a couple of very minor suggestions on the pdf. Knowing that PLOS does not provide author proofs I've also flagged possible typos.

6. PLOS authors have the option to publish the peer review history of their article (what does this mean?). If published, this will include your full peer review and any attached files.

Reviewer #1: **Yes: **Thilde Vildekilde

Reviewer #2: **Yes: **Dani Barrington

---

## [Author Response · Author response to Decision Letter 0]

28 Jun 2023

Please note we have also uploaded a response to reviewers for easier reading/formatting. 

Thank you for the continued consideration of our manuscript. We appreciate both reviewers’ positive feedback and emphasis of the value of this work.

R:We also appreciate their detailed suggestions of places to strengthen the reporting. We have revised the manuscript and responded to reviewer comments below.

Reviewer #1: 

Overall, my comments should be seen as minor (even though my writing is major:) ) and with small adjustments and amendments I recommend that this paper is published)

R:Thank you. We really appreciate your time in reviewing the manuscript and positive comments. We also appreciate the detailed feedback. We have revised the manuscript and provided responses below. 

Introduction:

- Reusable pads; I think it would strengthen the rationale for this study further if you bring in the sustainability aspect to the reusable trend (you mention it very briefly in the discussion). Your study actually provides a good argument for the need to see well managed reusable pads as a climate friendly alternative to disposable pads (in terms of related health risks).

R: This is a great point. We have been mindful of the word length of the paper, however, as you note our work addresses a broad set of research gaps, including around reusable products.

We have added the following to the introduction for greater emphasis of this issue:

The disposal of single-use menstrual products presents a significant challenge for waste management and global objectives for sustainable consumption (35). Reusable products can offer a waste and cost-effective alternative, however concerns of infection associated with inadequate washing and drying have attenuated enthusiasm for these products in low-and-middle-income country contexts. Research is needed to understand hygiene practices associated with reusable menstrual products and links to urogenital infection and irritation.

-I also think that you could introduce the rationale for focusing on MHM in the work place a bit more – its clear in your previously published papers (which I really enjoyed reading). Self-employed women working low paid jobs in somewhat sanitation and hygiene poor environments (markets) can be seen as particularly vulnerable and important to focus on. ‘MHM poverty’ as a concept could be used. Its interesting that you also include a different category of women (health providers and teachers) – but see my comment about these sub-groups below.

R: We have revised the introduction to further highlight the importance of workplaces within women’s lives, and for investigating linkages between hygiene practices and urogenital symptoms. We have added:

Limited research has described adult women’s menstrual practices, particularly in the workplace. As the importance of menstrual health has gained increased recognition, most focus in research and intervention has been for adolescents (36). Indeed, past research investigating linkages between hygiene practices and urogenital infection have focused on home-based behaviours or have not incorporated measures of workplace practices. Working women are likely to spend a significant among of time in their workplace, and these experiences and related needs require further exploration and understanding to be able to identify avenues for intervention (37). Understanding the menstrual practices of women in workplaces, alongside challenges for urination, which may increase risk of UTI (38), would allow a better understanding of potential causes of genital irritation in working women and inform improved support for menstruation in the workplace.

We have also revised the introduction under ‘the present study’ to expand upon the focus on women in markets, schools and health care facilities. 

Our study was undertaken as part of a broader exploration of women’s experiences of menstruation and sanitation in the workplace. This research focused primarily on women working in markets; a common worker group among women in Mukono with varied access to workplace water, sanitation and hygiene (WASH). Further we included a small proportion of teachers and health care facility workers; groups for which school and health care facility WASH is a recognised data and service gap (39). All three groups represent workplaces for which government has responsibility for service provision and would be able to implement changes in response to research findings.

We note that we had focused the introduction more on gaps in evidence for hygiene and RTI symptoms, and wanted to provide a meaningful overview of past literature on this association for readers. As you note our other manuscripts on women’s workplaces experiences focus much more heavily on this gap and need for research. However, we have added this additional emphasis.

- There is no descriptions of the general WASH standards in the market places. I can only imagine that very few toilets are serving huge numbers of women. And I can also imagine that the status of those WASH facilities are not very high. Same for access to water. Do markets have standpipes? Is water free and accesible?

R: We were not able to find any publicly available data on the market WASH infrastructure in Uganda. This means we are unable to present this in the introduction.

As part of our broader study we did undertake audits of all market sanitation facilities. However, reporting on this data goes far beyond the research question and focus of this manuscript. We feel it is best placed in a separate report that would have scope to unpack this data as would be needed.

In this study we report on women’s reports and their reactions to the WASH facilities; this is a closer correlate of their genital hygiene, so more relevant to this investigation.

METHODS:

- I just highly appreciate the break-down of the dichotomy of the reusable/non reusable category, in order to address the perceptions and classifications of reusable materials as being seen as ‘unhygienic’ per default. I think this point could also be further stressed in the papers discussion.

I have no expertise to evaluate the quality of your statistical methods.

R: Thank you. We really appreciate this acknowledgement, we also felt this comparison was an important contribution of this study.

We have dedicated a paragraph to this finding in the Discussion (paragraph 3), including stating that “These findings are particularly important in the context of increasing uptake of commercially produced reusable menstrual pads as a cost- and environmentally sustainable alternative to disposable pads. Our results suggest that, in this study setting, women taking up reusable clean pads experience the same benefits of lower prevalence of urogenital symptoms as those using disposables.”

RESULTS:

- 66% discussed the symptoms with a health provider; This is in my eyes a VERY high proportion – it would be SO interesting to know if they were prescribed antibiotics (which is often the standard treatment regime as you rightly mention in the discussion), since resistance against antibiotics for UTIs is a huge problem in other low income settings.

R:We agree! Unfortunately, we did not ask. A whole targeted study on women’s health care seeking for urogenital and menstrual symptoms, care provider perspectives and capacity, and prescribing behaviours is absolutely warranted (and needed to do this justice).

We have revised the first paragraph in the discussion to emphasise this point and to note the need for future research. We have stated:

“Among this group, two thirds (66%) discussed their symptoms with a health care provider. This finding alone suggests that most women found urogenital symptoms to cause sufficient discomfort and concern to seek care. Further research is needed to understand investigation and management of such symptoms, and quality of care received.”

- You are not mentioning the quite astonishing finding of the large proportion of women who report that they discard the pads in a sanitation facility! Being forced to discard menstrual materials in toilets is a sign that women do not have other suitable private places to manage their menstrual waste. It’s a huge environmental AND engineering problem that pads are discarded into toilets.

R: We have added this to the text of the results section under ‘Menstrual care at home and at work’ noting:

The greatest proportion of participants (43%, n=156) disposed of their used menstrual materials into the pit latrine or toilet in the workplace, while 29% (n=105) disposed into a waste bin or bucket at the sanitation facility, 25% (n=132) took materials home with them to dispose of.

We have also added this to the discussion:

“Further over four in ten (43%) of participants reported disposing of their used menstrual materials into the workplace sanitation facility, presenting risks to sanitation infrastructure and the environment (35).”

- No separate analysis for teachers or health providers is provided - why not? Its VERY relevant to get insights into status and progress of sanitation and MHM facilities and practices here, since sanitation in health facilities and in schools have been named as two major areas of intervention to support universal access to sanitation and to support MHM. Even though it may not be possible to arrive at statistically significant results for the two small sub-groups of respondents, I believe its relevant to see the results. I suspect that health personnel and teachers may have both better access to sanitation and MHM conditions AND lower frequencies of UTIs due to a high level of health literacy. These findings would lead to observations and discussions about the specific vulnerabilities of private sector, self-employed women, compared to women of higher education, in secure and public sector jobs – there is hardly ANY evidence in this field, so these results would be highly appreciated.

R: As the reviewer notes the sample size for comparisons of health care providers and teachers alone would be very small. We also note that our analyses adjust for worker type, so lower sanitation access and greater potential UTI/RTI symptoms among some worker groups (e.g., those in markets) is included in the adjusted estimates. It would then make it very difficult to compare separated analyses (with insufficient sample sizes) to these estimates.

Because the sample size (particularly once broken down by experiencing symptoms or material type) would be very small, we feel this would lend itself to the percentages being misinterpreted. 

In the discussion we have expanded this note:

“Due to practical limitations, only a small number of teachers and healthcare facility workers were included. Multivariable comparisons included adjustment for workplace type, poverty, and education level, however, results mostly describe the experiences and associations between practices and urogenital symptoms among women working in markets. Future research is needed to capture the sufficiency of workplace WASH services across worker types in different contexts.”

DISCUSSION:

- There is no mentioning or discussion of the descriptive findings of the general lack of access to suitable sanitation and hygiene facilities at the workplace. As a minimum I would like if you could refer to your other papers covering this in more detail. MHM in the workplace is still a hugely under-researched area within MHM research – especially in low income settings. I am well aware that this is not the purpose of this paper, but all findings that shed light on the MHM conditions, situation and problems at women’s workplaces, are relevant to report.

R: We have revised the discussion to incorporate this.

We have added a new paragraph stating:

“Across our sample, many women struggled with menstrual care and sanitation access in their workplace. A total 61% of participants reported needing to delay urinating at work in the past month, and over a third (35%) were not able to change their menstrual materials when they wanted to. Further over four in ten (43%) of participants reported disposing of their used menstrual materials into the workplace sanitation facility, presenting risks to sanitation infrastructure and the environment (35). Women’s menstrual experiences in the workplace have been neglected (55), with only more recent attention highlighting the many challenges faced (14, 56, 57). More research and action are needed to understand women’s menstrual needs at work and identify effective strategies for support.”

- Linked to my comment under results: I think that the discussion could be improved by adding some reflections about your informant group; Who do they represent? They are a special kind of women, and they work in special types of work places with special types of sanitation and hygiene standards/situations. To which groups and settings can your findings be translated? Hundreds of thousands of women in other low income settings are self-employed, growing some produce at home, selling it at markets, or working as suppliers, middle-men, brokers, and storeowners in similar markets all over the world. I know of similar groups of women in urban Ghana, rural Togo, urban Vietnam, rural and urban Sri Lanka. How are your results relevant beyond the specific ‘Ugandan markets’ context?

R: We have revised the Introduction of the paper to expand further on our selection of market, teacher and health care facility workers

Our study was undertaken as part of a broader exploration of women’s experiences of menstruation and sanitation in the workplace. This research focused primarily on women working in markets; a common worker group among women in Mukono with varied access to workplace water, sanitation and hygiene (WASH). Further we included a small proportion of teachers and health care facility workers; groups for which school and health care facility WASH is a recognised data and service gap (39). All three groups represent workplaces for which government has responsibility for service provision and would be able to implement changes in response to research findings.

Our study reports primarily on the association between menstrual hygiene practices and urogenital symptoms. This association is relevant across all groups, and the biological mechanisms of infection are not specific to context. However, the extent to which a population is likely to be exposed to unhygienic practices, how severe the lack of facilities or pathogens in within sanitation or water infrastructure are likely to differ. 

We have reported on the types of materials used, the type of sanitation facility access, and the sociodemographic characteristics of our sample. We have also adjusted analyses for these variables. In providing this information we believe there is sufficient contextual information for readers to determine the applicability of results to their own circumstances. It is not feasible for us to identify specific other comparable contexts – as you have highlighted there are certainly many women in low-resource settings that are likely to face similar challenges in accessing changing facilities, affording menstrual products, and enacting cleaning procedures. We believe it is best for those using the research to assess comparability relevant to their own context.

- Social desirability bias: Specifically, I think you can hypothesise that women may over report the practice of; drying materials fully in the sun and covering drying materials (two messages which are often heard in MHM promotion messages), and using improved sanitation facilities when at work and bringing water with them for washing when changing. Instead of merely mentioning potential bias, I think you should critically reflect on them, using some of your vast contextual knowledge about Ugandan market places and their structural conditions; Is it really realistic that women can access and afford to use improved sanitation facilities near the markets every time they need to change menstrual materials? From my own experience in urban Ghana, this is not very likely. Long queues, high costs, VERY poor hygienic conditions, and no water or wiping materials are just some of the reasons that women do not choose to use facilities. Also, accessing an improved sanitation facility, does NOT mean that its hygienic and has lower levels of infection risks to the women; fecal materials on the floor and walls, full pits, very strong odors and maggots are just some of the problems reported in many public sanitation facilities in low income settings.

R: In the Strengths and Limitations we have stated:

“Answers to questions about menstrual and urinary management and urogenital symptoms are subject to recall and social desirability bias, particularly if participants were embarrassed to report menstrual care practices like reusing unclean materials and infrequent handwashing experiencing urogenital symptoms in interviews. Thus, participants may have underreported less hygienic and may also have underreported symptoms. This may have dampened the potential effect size.”

We have stated the effect that social desirability is likely to have on the findings: attenuating the identified effect sizes. As a Discussion section, we need to ensure that what we discuss is what has been reported in the results of the paper. We do state that we expect that women underreported less hygienic practices, but beyond this we are not comfortable drawing further assumptions about the extent to which this was under reported. As noted above, we would need a separate report to do justice to describing in full the sanitation infrastructure in workplaces and this study focused on associations between hygiene practices and urogenital symptoms. 

CONCLUSION:

I find that the text here becomes rather generic. I would like to see you specifically refer to your study context (markets, low income setting) and the target groups you have investigated. The conclusion about “Reliable access to soap and clean water and private spaces to adequately clean and dry reusable materials is required” is so unspecific, that its almost true for all of us, always. Focus on the workplace. That’s were your study is new and relevant.

R: We have revised the conclusions to note the specifics of the sample. We have also expanded upon the need for menstrual-friendly sanitation infrastructure in the specific workplaces assessed. 

We note that cleaning practices for menstrual absorbents are likely to happen at home, and this was a significant part of this piece of research. So, while our work is novel in integrating changing practices in the workplace and the influence of the workplace sanitation facility, it is also cleaning practices (enacted at home) that we focus on.

Reviewer #2: 

Thank you for the opportunity to review this interesting manuscript. The authors provide new insights on the links between menstrual hygiene and sanitation types alongside the use of clean/unclean disposable/reusable menstrual/improvised materials with urogenital symptoms. It is particularly useful that they have differentiated between reusable pads and improvised materials.

The authors describe a robust study and thoroughly investigates the limitations of the work. I have made a couple of very minor suggestions on the pdf. Knowing that PLOS does not provide author proofs I've also flagged possible typos.

R: We really appreciate the reviewer’s time in peer-reviewing the manuscript. Thank you for your positive feedback. 

We also really appreciate the time taken to flag possible typos and formatting inconsistencies. We have revised these throughout based on your PDF comments, and an additional review by our team. (We have not listed typos/formatting revisions below)

Other comments from pdf file 

I've been reading on trans reproductive health recently, particularly around language (from both sides of the inclusive language divide), and I think it would be useful to preface this with 'other' - as how it is currently could be seen by some as not considering women as people.

R: Thank you, this is helpful feedback. ‘Other people who menstruate’ is also more aligned with the wording of the definition. We have revised accordingly. 

Unclear why this was done alongside the marketplaces. [primary schools and health care facilities]

R: We have added further justification to the introduction capturing our reasons for focusing on markets, schools and health care facilities.

We have revised this sentence of the methods to note that the neighbouring schools/health care facilities were sampled to support study feasibility.

can we have an idea of sizes of the markets; at least how much bigger the 'largest' market is?

R: Thanks for this feedback, a good point. We have added the following sentence in the ‘Recruitment’ section

Estimated female worker population of most markets ranged from 10-150, with one large central market hosting an estimated 950 female workers.

---

## [Decision Letter · Decision Letter 1]

7 Jul 2023

The association between menstrual hygiene, workplace sanitation practices and self-reported urogenital symptoms in a cross-sectional survey of women working in Mukono District, Uganda

PONE-D-23-10980R1

Dear Dr. Hennegan,

We’re pleased to inform you that your manuscript has been judged scientifically suitable for publication and will be formally accepted for publication once it meets all outstanding technical requirements.

Kind regards,

Alison Parker

Academic Editor

PLOS ONE

Additional Editor Comments (optional):

Reviewers' comments:

Reviewer's Responses to Questions

**Comments to the Author**

1. If the authors have adequately addressed your comments raised in a previous round of review and you feel that this manuscript is now acceptable for publication, you may indicate that here to bypass the “Comments to the Author” section, enter your conflict of interest statement in the “Confidential to Editor” section, and submit your "Accept" recommendation.

Reviewer #1: All comments have been addressed

Reviewer #2: All comments have been addressed

2. Is the manuscript technically sound, and do the data support the conclusions?

Reviewer #1: Yes

Reviewer #2: Yes

3. Has the statistical analysis been performed appropriately and rigorously? 

Reviewer #1: I Don't Know

Reviewer #2: Yes

4. Have the authors made all data underlying the findings in their manuscript fully available?

Reviewer #1: Yes

Reviewer #2: Yes

5. Is the manuscript presented in an intelligible fashion and written in standard English?

Reviewer #1: Yes

Reviewer #2: Yes

6. Review Comments to the Author

Reviewer #1: (No Response)

Reviewer #2: The authors have addressed the reviewer comments and I believe this paper should be accepted for publication.

7. PLOS authors have the option to publish the peer review history of their article (what does this mean?). If published, this will include your full peer review and any attached files.

Reviewer #1: No

Reviewer #2: **Yes: **Dani Barrington

---

## [Editor Report · Acceptance letter]

12 Jul 2023

PONE-D-23-10980R1 

The association between menstrual hygiene, workplace sanitation practices and self-reported urogenital symptoms in a cross-sectional survey of women working in Mukono District, Uganda 

Dear Dr. Hennegan:

I'm pleased to inform you that your manuscript has been deemed suitable for publication in PLOS ONE. Congratulations! Your manuscript is now with our production department. 

Kind regards, 

on behalf of

Dr. Alison Parker 

Academic Editor

PLOS ONE